# Comparative Metabolome and Transcriptome Analysis of Anthocyanin Biosynthesis in White and Pink Petals of Cotton (*Gossypium hirsutum* L.)

**DOI:** 10.3390/ijms231710137

**Published:** 2022-09-04

**Authors:** Dongnan Shao, Qian Liang, Xuefeng Wang, Qian-Hao Zhu, Feng Liu, Yanjun Li, Xinyu Zhang, Yonglin Yang, Jie Sun, Fei Xue

**Affiliations:** 1Key Laboratory of Oasis Eco-Agriculture, College of Agriculture, Shihezi University, Shihezi 832003, China; 2CSIRO Agriculture and Food, GPO Box 1700, Canberra 2601, Australia; 3Cotton Research Institute, Shihezi Academy of Agriculture Science, Shihezi 832000, China

**Keywords:** *Gossypium hirsutum* L., petal, coloration, anthocyanins, RNA-seq, metabolome

## Abstract

Upland cotton (*Gossypium hirsutum* L.) is one of the important fiber crops. Cotton flowers usually appear white (or cream-colored) without colored spots at the petal base, and turn pink on the next day after flowering. In this study, using a mutant showing pink petals with crimson spots at their base, we conducted comparative metabolome and transcriptome analyses to investigate the molecular mechanism of coloration in cotton flowers. Metabolic profiling showed that cyanidin-3-O-glucoside and glycosidic derivatives of pelargonidins and peonidins are the main pigments responsible for the coloration of the pink petals of the mutant. A total of 2443 genes differentially expressed (DEGs) between the white and pink petals were identified by RNA-sequencing. Many DEGs are structural genes and regulatory genes of the anthocyanin biosynthesis pathway. Among them, *MYB21*, *UGT88F3*, *GSTF12*, and *VPS32.3* showed significant association with the accumulation of cyanidin-3-O-glucoside in the pink petals. Taken together, our study preliminarily revealed the metabolites responsible for the pink petals and the key genes regulating the biosynthesis and accumulation of anthocyanins in the pink petals. The results provide new insights into the biochemical and molecular mechanism underlying anthocyanin biosynthesis in upland cotton.

## 1. Introduction

As one of the visible traits of plants, color is of great significance to the growth and development of plants. Many studies have shown that carotenoids, flavonoids, and betalains are the main pigments in plant color formation [1]. Particularly, flavonoids have attracted more attention because they are widely distributed secondary metabolites in plants, and contribute to tissue coloring, plant environmental adaptation, fruit development, and even human health [2]. Anthocyanins are water-soluble pigments belonging to the flavonoid family that mainly contribute to the red/pink/blue/purple coloration of plants, often forming complex color patterns, such as spots, stripes, and veins, in different organs of plants, particularly in flowers [1,3,4]. Flower coloration is one of the most attractive traits of ornamental plants, mainly determined by the type and content of six different anthocyanidins. Peonidin and cyanidin contribute to colors from pink to red; pelargonin contributes to brick red; and delphinidin, malvidin, and petunidin contribute to the colors between purple and blue [5,6,7].

To date, anthocyanin biosynthesis, one branch of the flavonoid biosynthesis pathway, has been almost completely elucidated in many plant species, such as arabidopsis (*Arabidopsis thaliana*), maize (*Zea mays*), apple (*Malus domestica*), tomato (*Solanum lycopersicum*), grapevine (*Vitis vinifera*) and potato (*Solanum tuberosum*) [8,9,10,11]. Firstly, phenylalanine, as the precursor of anthocyanin biosynthesis, is catalyzed by phenylalanine ammonia lyase (PAL), cinnamate 4-hydroxylase (C4H), and 4-coumarin: Co-A ligase (4CL) to form Coumaric acid coenzyme A. Secondly, one molecule of coumaric acid and three molecules of malonyl-CoA produce colorless anthocyanins through several reactions catalyzed by enzymes such as chalcone synthase (CHS), chalcone isomerase (CHI), flavanone 3-hydroxylase (F3H), flavonoid 3′-hydroxylase (F3′H), flavonoid 3′,5′-hydroxylase (F3′5′H), and dihydroflavanol-4 reductase (DFR). Then, colorless anthocyanins are catalyzed by anthocyanin synthase/leucocyanidin dioxygenase (ANS/LDOX) and UDP-glucose:flavonoid-*O*-glycosyltransferase (UFGT) to form a stably colored anthocyanin [12,13]. Finally, the stable anthocyanins are transported into vacuoles, causing the corresponding tissues to display color. The mechanism of anthocyanin transport is not completely clear, but three models have been proposed for vacuolar transport of anthocyanins. (1) With the assistance of glutathione S-transferase (GST), anthocyanins are targeted and located to near vacuoles, and then recognized by the C-type of ABC (ABCC) transporters on vacuoles and transported across the membranes into vacuoles; (2) Multidrug and toxic compound extrusion (MATE) transporters on the vacuole transport anthocyanins across the membrane to the vacuole. This process requires an H^+^ concentration gradient generated by H^+^-ATPase proton pump; (3) Anthocyanins are encapsulated by anthocyanin vacuole inclusions (AVIs) and enter into vacuoles by means of membrane fusion [14,15]. The vesicular transportation is mediated by SNARE (soluble *N*-ethylmaleimide-sensitive fusion protein attachment protein receptors) protein complexes, which are considered to be involved in the cellular transport in higher plants under stress responses [16,17]. Besides the structural genes mentioned above, anthocyanin biosynthesis is also controlled by regulatory genes. In recent decades, studies have confirmed that the expression of the structural genes mentioned above has different degrees of synergy, which is directly controlled by the MBW complex formed by MYB, bHLH, and WDR transcription factors (TFs) [13,18,19]. In addition to the MBW complex, several TFs, such as DELLA, JAZ, COP1, PIF3, SPL, WRKY, B-box protein, and bZIP, have been reported to regulate anthocyanin biosynthesis via interaction with the MBW complex, or acting on the upstream of the MBW complex [20,21,22,23,24,25,26,27,28,29,30]. In addition, some external environmental factors can also induce anthocyanin biosynthesis, such as light, temperature, and drought [22,31,32,33,34,35,36].

Cotton (*Gossypium* spp.), as a crop of the Malvaceae family, has many phenotypic polymorphisms, which have been used in modern genetic studies. Among them, flower color has attracted much attention as an obvious trait. More than a century of research has shown that the color of cotton flowers is the result of the joint action of flavonols and anthocyanins [37]. Although many different flower colors have been observed in the genus *Gossypium*, the main flower color of the cultivated cotton *G*. *hirsutum* and *G*. *barbadense* is white (or cream) and yellow, respectively [37,38]. To date, four genomic loci (*R1*, *R2*, *Rs*, and *Rd*) controlling anthocyanin accumulation have been identified in Upland cotton [39,40,41]. The genes of the *R1* and *Rs* loci contributing red color to cotton plants are a pair of homeologous MYB transcription factors *GhPAP1D*/*GhRLC1* and *GhPAP1A*, respectively [40,42]. *Rd* has a dwarf phenotype compared with *R1* and *Rs* [43]. *R2* is associated with the red petal spots in green leaf cotton [41,43]. In addition, *GbBM* (a MYB TF) causes purple spots on the bottom of the petals of *G. barbadense* by activating anthocyanin accumulation [44].

We found a natural mutant, designated *Pink Flower* (*PF*), showing pink flowers with crimson spots at the bases of petals (Figure 1A,B and Appendix A). The *PF* mutant provides a resource for exploring the mechanism of anthocyanin biosynthesis and accumulation in upland cotton. To elucidate the changes in metabolites and transcriptional regulation of the anthocyanin biosynthetic pathways in the *PF* mutant, using the white flower cotton cultivar X74 as a control, we compared their metabolome and transcriptome. We identified metabolites unique to the pink petals and uncovered the key structural genes and regulatory factors associated with the biosynthesis and accumulation of anthocyanins in the pink petals.

## 2. Results

### 2.1. Analysis of Anthocyanin Composition in Petals of X74 and PF

The total contents of anthocyanins in the petals of *PF* (PA and PB) and X74 (WA and WB) (Figure 1A–D) were measured by a spectrophotometer. The total anthocyanins were significantly higher in PA and PB, particularly in PA, compared to WA and WB, in which only a very small amount was detected (Figure 1E). To further explore the metabolic mechanism of the *PF* flowers, we analyzed the anthocyanin metabolites in PA, PB, WA, and WB (Figure 2C). In total, 38 anthocyanin metabolites were identified (Appendix A), and 9 anthocyanin metabolites are unique to the pink petals (Figure 2B). In addition, 4 unique anthocyanin metabolites were found in the spot region (PA) of the pink petals compared with the non-spot region (PB), but their concentrations are relatively low. More anthocyanin metabolites were detected in PA (36) and PB (33) than in WA (23) and WB (24). Quercetin-3-O-glucoside and delphinidin are the main differential metabolites between the spotted (PA and WA) and non-spotted (PB and WB) petal regions. The main difference between pink (PA and PB) and white (WA and WB) petals is in cyanidin and procyanidin metabolites. It is worth noting that the content of cyanidin-3-O-glucoside in PA and PB is 743- and 310-fold higher than that in WA and WB, respectively, and that the concentration of cyanidin-3-O-glucoside is higher in PA than in PB (Figure 2C and Appendix A), indicating that cyanidin-3-O-glucoside might be crucial for the pink color formation.

### 2.2. Overview of the RNA Sequencing Data

To explore the molecular mechanism related to the pink coloration of the *PF* petals, we performed RNA-seq using samples collected from the WA, WB, PA, and PB regions (Figure 1B,D and Figure 2A). Clean reads were obtained after filtering out low-quality reads. The Q20 and Q30 of each sample were greater than 97.4% and 92.63%, respectively. The GC content of the reads was 42.97–43.42%. Between 96.54% and 97.04% of the clean reads were mapped to the upland cotton genome (*Gossypium hirsutum*, ZJU-improved_v2.1_a1). Among the mapped reads, the uniquely mapped reads accounted for 92.06–92.95% (Appendix A). The Pearson correlation coefficient and principal component analysis of the samples based on the FPKM (fragments per kilobase of transcript per million mapped reads) values showed that the biological replicates exhibited similar expression patterns, indicating the high reliability of the sequencing data (Appendix A). To further confirm the accuracy of the transcriptomic data, we selected 11 differentially expressed genes of the anthocyanin biosynthesis pathway for qRT-PCR verification. Although there were slight differences between the results of the RNA-seq and qRT-PCR, the overall trend was consistent (Appendix A).

### 2.3. Analysis of Functional Enrichment of Differentially Expressed Genes

To identify the DEGs (based on FDR ≤ 0.05 and fold change ≥ 2) involved in *PF* flower coloration, based on FPKM values, we did four pairwise comparisons. A total of 9243, 7640, 1719, and 1642 DEGs were identified in the pairwise comparison of PB_vs._PA, WB_vs._WA, WA_vs._PA, and WB_vs._PB, respectively. Interestingly, the number of DEGs between the spotted and non-spotted regions of petals (PB_vs._PA and WB_vs._WA) was much greater than that between the same part of petals of different genotypes (WA_vs._PA and WB_vs._PB) (Figure 3A). A Venn-diagram illustrates that 5395 and 669 genes were unique in different petal tissues of the same genotype (PB_vs._PA and WB_vs._WA) and the same petal tissue of different genotypes (WA_vs._PA and WB_vs._PB), respectively (Figure 3B). Accordingly, hierarchical clustering analysis also showed a clear difference in the expression patterns of the DEGs between different genotypes and tissues (Figure 3C).

We assigned all the DEGs of different comparison groups to three major GO categories, including cellular components (CC), molecular functions (MF), and biological processes (BP) (Appendix A). The top CC sub-categories in WB_vs._WA and PB_vs._PA were ‘integral component of plasma membrane’, ‘kinesin complex’, and ‘extracellular region part’. The DEGs in WA_vs._PA and WB_vs._PB were not significantly enriched in CC terms (Figure 4A and Appendix A). Of the MF categories, ‘DNA/RNA polymerase activity’, ‘endonuclease activity’, and ‘ribonuclease activity’ were found to be enriched in WA_vs._PA and WB_vs._PB. The DEGs from WB_vs._WA and PB_vs._PA were enriched with ‘CoA synthase activity’, ‘transporter activity’, and ‘glycosyl transferase activity’. The BP terms of ‘DNA integration’, ‘histone ubiquitination’, and ‘regulation of cell division’ were enriched in WA_vs._PA and WB_vs._PB comparisons (Figure 4A and Appendix A). The GO terms ‘cuticle development’, ‘phenylpropanoid biosynthetic process’, and ‘drug transmembrane transport’ were the top ones in WB_vs._WA and PB_vs._PA comparisons. In addition, genes associated with sugar metabolic, starch metabolic, lignin, flavonoid, and sterol biosynthetic process were also found to be enriched in WB_vs._WA and PB_vs._PA (Appendix A).

All the DEGs from different comparison groups were further subjected to KEGG pathway enrichment analysis (Appendix A). It showed that a high number of genes were associated with ‘Metabolic pathways’ and ‘Secondary metabolite biosynthesis’, followed by ‘Plant hormone signal transduction’, ‘MAPK signaling pathway—plant’ and ‘carbohydrate metabolism’, including ‘Carbon metabolism’, ‘Starch and sucrose metabolism’, and ‘Fructose and mannose metabolism’ (Figure 4B and Appendix A). In addition, the DEGs identified in PB_vs._PA (spot vs. non-spot region) was significantly enriched with the pathways of ‘fatty acid elongation’, ‘ABC transporters’ and ‘Glutathione metabolism‘.

Traditional strategies for gene expression analysis have focused on identifying individual genes that exhibit differences between two states of interest. Moreover, most determine whether a group of differentially expressed genes is enriched for a pathway or ontology term (such as GO and KEGG) by using overlap statistics such as the cumulative hypergeometric distribution. Although useful, due to the defect of threshold screening of GO and KEGG enrichment analysis, some genes with insignificant expression difference but important biological significance will be ignored. By contrast, GSEA considers all the genes in the experiment, not only those genes that were above any cutoff in terms of fold-change or significance. We thus further investigated the key genes correlated with flower pigmentation by GSEA (Gene Set Enrichment Analysis) using a KEGG-based list (196 gene sets). The up-regulated gene sets from all comparisons are provided in Appendix A. Many of the top 20 pathways from each comparison overlapped between PB_vs._PA and WB_vs._WA as well as between WA_vs._PA and WB_vs._PB (Figure 5A). Consistent with the KEGG enrichment analysis results, ‘Glycolysis/Gluconeogenesis’, ‘Pentose phosphate pathway’, ‘Fructose and mannose metabolism’, ‘MAPK signaling pathway—plant’, and ‘Plant hormone signal transduction’ were mainly related to the higher expression gene sets (Figure 4B and Figure 5B–E). In addition, the ‘Fatty acid elongation’, ‘ABC transporters’, and ‘Glutathione metabolism’ pathways were significantly enriched in KEGG analysis did not appear in the GSEA analysis. By contrast, ‘Anthocyanin biosynthesis’ in PB_vs._PA and WB_vs._WA and ‘Plant—pathogen interaction’ in WA_vs._PA and WB_vs._PB were only found in the GSEA analysis (Figure 5B–E).

### 2.4. Analysis of DEGs Related to Biosynthesis and Transport of Anthocyanidins

Given the increase in the content of anthocyanins in pink flowers of the *PF* mutant, we conducted a detailed analysis of the genes involved in the phenylpropanoid, flavonoid, flavonol and flavone, and anthocyanidin biosynthesis pathways. A total of 80 DEGs potentially involved in anthocyanin biosynthesis were found, including *GhPAL*, *Gh4CL*, *GhCHS*, *GhCHI*, *GhFLS*, *GhF3H*, *GhF3′H*, *GhF3′5′H*, *GhANS*, *GhLAR*, *GhANR*, and *GhUFGTs* (Figure 6 and Appendix A). The PCC (Pearson’s correlation coefficient) between the expression level of these DEGs and the DEMs (differentially expressed metabolites) was calculated. It showed that Cyanidin-3-O-glucoside biosynthesis was positively and negatively regulated by 12 and 7 DEGs, respectively (|PCC| ≥ 0.6, Appendix A). The 12 positive regulators included 7 *UGT* (UDP-glycosyltransferase) genes, 2 *PAL* genes, 2 *LAR* genes, and 1 *ANR* gene. Among the 7 *UGTs*, the expression level of *GH_D11G3364*, namely *GhUGT88F3*, showed a significant positive correlation with the Cyanidin-3-O-glucoside content (PCC > 0.9, Appendix A). Some metabolites (such as Cyanidin-3-O-galactoside, Pelargonidin-3-O-glucoside, and Peonidin-3-O-galactoside) with the same concentration change trend as Cyanidin-3-O-glucoside might also be involved in the formation of pink color of petals (Appendix A).

After synthesis, anthocyanins are transported from the cytosol to the vacuole for storage, which is a key step in the coloration of tissues or organs. We identified four genes annotated as *GSTF* from the ‘Glutathione metabolism’ pathway in KEGG analysis (Appendix A). Notably, we found that *GhGSTF12* (*GH_A07G0814* and *GH_D07G0816*), previously reported to be involved in anthocyanin accumulation in cotton [41], was one of the four genes. We identified many differentially expressed *ABC* (ATP-binding cassette) genes, such as *ABCA* (1 genes), *ABCB* (17 genes), *ABCC* (14 genes), and *ABCE* (34 genes), most of which were highly up-regulated in WA and PA (Appendix A). We also identified some Multidrug and toxic compound extrusion proteins (*MATEs*) and Vesicle-associated membrane proteins (*SNAREs*) from DEGs significantly enriched in ‘drug transmembrane transport’ and ‘SNARE binding’/‘SNARE complex’, respectively (Appendix A). Among these transporters, the expression level of *GhGSTF12* (GH_A07G0814 and GH_D07G0816), *GhABCC12* (GH_A12G0446), and *GhABCC4* (GH_A12G1481) showed a significant positive correlation with the Cyanidin-3-O-glucoside content in different samples (PCC > 0.9, Appendix A).

Transcription factors play an essential role in anthocyanin biosynthesis and transport. Of all the DEGs (12,516 genes from all four comparisons), a total of 1048 genes were identified as putative TFs and regulators. They were categorized into 76 TF families (Appendix A). Among them, the majority belongs to the AP2/ERF-ERF family, followed by bHLH, MYB, bZIP, C2H2, NAC, HB-HD-ZIP, WRKY, and GRAS families (Appendix A). By calculating the Pearson’s correlation coefficient (PCC) between the FPKM of these TFs, and the content of Cyanidin-3-O-glucoside, 25 key TFs (|PCC| ≥ 0.9) with a potential role in flower coloration were identified, including 14 negative regulators and 11 positive regulators (Appendix A). The 11 positive regulators related to accumulation of Cyanidin-3-O-glucoside include one MYB, one C3H, two ZIP, one B3, one IWS1, one GARP-ARR-B, one OFP, one C2C2-Dof, and two AP2/ERF genes (Appendix A). Interestingly, MYB21 (GH_D12G1622), the only MYB among the positive regulators, is not one of those (GH_A07G0850 and GH_D07G0852) that have been reported to be the responsible genes for the red leaf phenotype of *Rs* and *R1* [40,42].

### 2.5. Co-Expression Analysis of the Genes Related to the Anthocyanidin Biosynthesis Pathway

In order to investigate the gene network regulating anthocyanin metabolism in the *PF* mutant, the 5940 non-redundant DEGs were subjected to WGCNA. These DEGs were clustered into eight modules (Figure 7A,B). Seven representative anthocyanin metabolites were used as traits in the gene module–trait relationship analysis. The brown module had a significant positive correlation with the content of five metabolites, including Cyanidin-3-O-glucoside, Pelargonidin-3-O-glucoside, Peonidin-3-O-galactoside, Procyanidin B2, and Procyanidin C1, whereas the yellow module was significantly negatively correlated with these five metabolites (Figure 7B). Delphinidin-3-O-glucoside and Quercetin-3-O-glucoside were significantly positively correlated with the turquoise module, but negatively correlated with the green, blue, and black modules (Figure 7B). The blue and green modules showed a positive relationship with Cyanidin 3-O-glucoside, Pelargonidin-3-O-glucoside, and Peonidin-3-O-galactoside (Figure 7B).

In order to find the key regulatory genes from the turquoise, yellow, brown, and green modules, we filtered out the hub genes from these modules based on PCC between individual genes and the content of the seven representative metabolites. The 78 hub genes selected based on |PCC| ≥ 0.95 were constructed into two regulatory networks, each associated with two or four of the seven metabolites, and no gene was associated with Peonidin-3-O-galactoside (Figure 7C). Delphinidin-3-O-glucoside and Cyanidin-3-O-glucoside seemed to be regulated by more genes than Quercetin-3-O-glucoside, Pelargonidin-3-O-glucoside, Procyanidin B2, and Procyanidin C1. Among these regulatory genes, *GhGSTF12* (GH_D07G0816) was significantly positively correlated with Cyanidin-3-O-glucoside and Pelargonidin-3-O-glucoside, while *GhUGT88F3* (GH_D11G3364) and *GhVPS32.2* (GH_A13G2237) were positively correlated only with Cyanidin-3-O-glucoside, the main contributor to the pink petal phenotype (Figure 6 and Figure 7C).

## 3. Discussion

### 3.1. Cyanidin-3-O-Glucoside Is the Main Contributor of the Pink Color Observed in the PF Mutant

For most plants, flavonoids, especially anthocyanins, are the main pigments in the coloration of their organs or tissues. For example, the colors seen on the skin of citrus, peach, pear, and eggplant [26,35,45,46], the flesh of apples, strawberries, and tomatoes [47,48,49], and the flowers of various ornamental plants such as petunia, peony, and chrysanthemum [50,51,52] are all resulted from tissue-specific accumulation of anthocyanins. Studies have shown that the primary shade of flower color (from red to blue violet) is mainly determined by six main anthocyanin metabolites (cyanidins, delphinidins, pelargonins, peonidins, malvidins, and petunidins) [53]. But, the proportion of anthocyanin metabolites and their biosynthetic genes can be significantly different in different plant species, so it is necessary to conduct personalized analysis. In this study, we found that the total anthocyanin content of petals was much higher in the *PF* mutant than in X74, particularly in the colored spots observed at the petal base (Figure 1E). A total of 38 different anthocyanin metabolites were identified in the petals of *PF* and X74 (Figure 2). The anthocyanins often exist in the form of monoglycosides (occurring at the 3-position), probably to maintain stability. Consistent with previous studies, quercetin, as one of the background colors of upland cotton petals, is the most abundant flavonoid metabolite in both white and pink petals [37]. Of the six major anthocyanin metabolites, cyanidins, delphinidins, pelargonins, and peonidins were detected in *PF* and X74 petals (Appendix A). The content of Cyanidin-3-O-glucoside is significantly higher in the pink petals than in the white petals, and is considered to be the main contributor for the color variance of X74 and *PF* petals, consistent with previous studies in different plant species [44,52,53,54]. The white (or cream) flower of upland cotton turns red on the next day after flowering, which was also proved to be due to the sharp increase in cyanidins. This leads us to speculate that the pink petal color of *PF* is due to early activation of the Cyanidin-3-O-glucoside biosynthesis pathway, although whether it shares the activation mechanism involved in petal color change after flowering remains to be further studied. Furthermore, some monoglycoside derivatives of cyanidins, pelargonins, and peonidins (such as Cyanidin-3-O-sambubioside, Cyanidin-3-rutinoside, Peonidin-3-O-galactoside, and Pelargonidin-3-O-glucoside) were not found or were present in very low levels in the white petals compared with the pink petals, implying that these anthocyanins may play fine-tuning roles in deepening the petal color of *PF* (Figure 2). Interestingly, delphinidins has a high concentration in both white and pink petals, implying that delphinidins have little to do with the pink color of *PF* petals; this could be because: (1) delphinidins have not been transferred to vacuoles, an essential step for tissue coloration; and/or (2) the pH in the petal vacuole is not suitable for coloring by delphinidins.

### 3.2. Key Genes Responsible for Biosynthesis and Accumulation of Anthocyanins in PF Petals

As a branch of the flavonoid biosynthesis pathway, anthocyanin biosynthesis has been deeply studied in many plant species [55]. Consistent with the published results, many structural genes involved in anthocyanin biosynthesis, such as *PAL*, *C4H*, *4CL*, *CHS*, *CHI*, *F3H*, *FLS*, *F3′H*, *F3′5′H*, *ANS*, and *UFGT*, were differentially expressed between the white petals and the pink petals. Interestingly, we found that the expression level of most structural genes was the highest in PB, followed by WB, PA, and WA, implying that they are related to the biosynthesis of quercetin and delphinidins (Figure 6 and Appendix A). *DFR*, the key gene in the anthocyanin synthesis pathway, was highly expressed in both white and pink petals but is not a DEG, and may contribute to accumulation of cyanidins, pelargonins, peonidins, and delphinidins in both white and pink flowers. In addition, the 78 genes selected from WGCNA were mainly associated with Cyanidin-3-O-glucoside and Delphinidin-3-O-glucoside (Figure 7C). The gene network and association analysis also suggest that the formation of pink petals is due to the activation of Cyanidin-3-O-glucoside synthesis pathway. In the Cyanidin-3-O-glucoside regulatory network, *GSTF12*, *UGT88F3**,* and *VPS32.3* were highly positively correlated with Cyanidin-3-O-glucoside. Among them, the expression level of *GSTF12*, which has been confirmed by our previous study to be involved in the accumulation of anthocyanins in upland cotton, is consistent with the color difference of white and pink petals [41]. As the last structural gene of anthocyanin biosynthesis, UGTs (UFGTs or 3GTs) are responsible for catalyzing the glycosylation of anthocyanins and causing the diversity of anthocyanin metabolites in *Arabidopsis* [56]. The expression trend of *UGT88F3* in petal tissues suggests that it may be responsible for the glycosylation of Cyanidin-3-O-glucoside in pink petals, even if its expression level is not high. Furthermore, *VPS32.3*, described as vacuolar protein sorting-associated protein, may play an important role in transport of anthocyanins according to the AVI model [12,14].

Anthocyanin biosynthesis is directly regulated by the MBW complex composed of R2R3-MYB, bHLH, and WD40 [13]. A high number of DEGs were found to be MYB and bHLH genes (Appendix A and Appendix A). Based on the integration analysis of transcriptome and metabolome, the expression level of *MYB21* (GH_D12G1622) showed a trend consistent with the content of Cyanidin-3-O-glucoside in the white and pink petals (Appendix A and Appendix A). Conversely, the expression level of two bHLH (GH_A09G2156 and GH_A13G0532) genes in white petals was higher than in pink petals, which was contrary to the total anthocyanin content in white and pink petals (Appendix A and Appendix A). Whether the two bHLHs interact with MYB21 to regulate anthocyanin synthesis in pink petals remains to be further verified. While *GhPAP1A* (GH_A07G0850) and *GhPAP1D* (GH_D07G0852) have been previously identified as the key MYB TFs activating anthocyanin biosynthesis in red leaf mutants *R1* and *Rs* [40,42], both genes were not differentially expressed between the white and pink petals, although their expression levels in the non-spotted regions of both white and pink petals were much higher than that in the spotted regions (Appendix A), which leads us to speculate that the high expression level of most structural genes involved in anthocyanin biosynthesis in the non-spotted regions of the petal may be a result of the high expression of *GhPAP1A* or *GhPAP1D* in the non-spotted petal regions (Figure 6).

### 3.3. Outlook of Anthocyanin Biosynthesis in Upland Cotton

Cotton fiber is the main source of natural fiber in the textile industry. Naturally Colored Cotton (NCC) has attracted much attention as an environment-friendly resource. The molecular mechanism of the biosynthesis and accumulation of pigments in NCC fiber is largely unknown, although previous studies have shown that the production of brown cotton and green cotton is related to proanthocyanidins (PAs) (or their derivatives) and caffeic acid (or their derivatives), respectively [57,58,59]. It is worth noting that the biosynthesis of PAs and anthocyanins shares most of the structural and regulatory genes [57,60]. Activation of the anthocyanin biosynthesis pathway is often accompanied by an increase in the expression level of the genes (such as *LAR* and *ANR*) related to proanthocyanidin biosynthesis (Figure 6), and the subsequent increase in proanthocyanidin content [61]. Therefore, the identification of genes related to the biosynthesis and accumulation of anthocyanin is of great significance to understanding the mechanism underlying cotton fiber coloration.

Although fiber is the main product of cotton, its yield is far lower than that of cotton flower [37]. Therefore, cotton flower, which is rich in flavonoids, sugars, and other metabolites, has great potential in many aspects. For example, compared with the artificially synthetic anthocyanins, natural anthocyanins with strong antioxidant and free radical-scavenging properties are more favored by consumers. Cotton petals rich in anthocyanins may thus become a new source of natural anthocyanins. In addition, using heterosis to improve yield is one of the major breeding strategies of cotton. The utilization of male sterility is undoubtedly an effective method to produce hybrid seeds to reduce the cost and ensure the purity of varieties. Success of large-scale hybrid seed production depends on the efficient and economic reproduction of male sterile female parents. Introducing pigment deposition-related genes into hemizygous colored maintainer lines by genetic engineering (including fertility restoration genes), and crossing such maintainer lines to male sterile lines can produce 100% male sterile offspring by removing the colored seeds or seedlings [62,63]. This has been applied in maize, rice, and tomato [64,65,66]. In addition, anthocyanins not only protect flowers against UV-light, but also form particular patterns (such as petal spots, stripes, and patches) of flower colors to attract pollinators and thereby improve female reproductivity [44,67]. Therefore, based on the preference of pollinators for specific flower colors or particular color patterns, it may be possible to guide pollinators to conduct ‘point-to-point’ cross-pollination by manipulating the petal color or pigment distribution in male sterile lines and maintainer lines, so as to further reduce the production cost of artificial pollination. We hope our results will provide a foundation for applying the above conceptions in cotton cultivation.

## 4. Materials and Methods

### 4.1. Plant Materials and Sampling

*Pink Flower* (*PF*, *Gossypium hirsutum* L.) is a natural mutant showing a pink flower with crimson spots at the base of the petals (Appendix A and Figure 1B). XinLuZao 74 (X74, *Gossypium hirsutum* L.) is a white flower cultivar. All plants used in this study were grown in the Teaching Experimental Farm of Shihezi University (Shihezi, China) under natural light.

We collected petal samples from the crimson spot (PA) and non-spot areas (PB) of the *PF* mutant on the day before flowering (Figure 1B and Figure 2A). The corresponding petal tissues (WA and WB) were sampled from the white flowers of X74 (Figure 1D and Figure 2A). After sampling, all materials were immediately frozen in liquid nitrogen and stored at −80 °C until used.

### 4.2. Extraction and Measurement of Total Anthocyanins

According to a previous study [41], flower tissues were weighed (0.1 g), ground into powder in liquid nitrogen, and resuspended in 1 mL acidic methanol (1% hydrochloric acid, *w*/*v*) at room temperature for 12 h in dark. After centrifugation at 12,000× *g* rpm for 10 min at 4 °C, 1 mL of the supernatant was added to 4 mL of acidic methanol. The absorbance of the solution was measured with a U-5100 UV/VIS spectrophotometer (Shimadzu, Kyoto, Japan) at 530 nm, 620 nm, and 650 nm. The anthocyanin content was calculated with the following formula: Q = OD_λ_/ε × V/m × 10^6^, OD_λ_ = (A530 − A620) − 0.1 × (A650 − A620). [V (mL): Total volume of extract; m (g): Fresh weight of sample; ε: Molar extinction coefficient of anthocyanins 4.62 × 10^6^].

### 4.3. Qualitative and Quantitative Analysis of Anthocyanins

The quantitative analysis of anthocyanins was performed by Metware Biotechnology Co., Ltd. (Wuhan, China) based on the AB Sciex QTRAP 6500 LC-MS/MS platform. The petal samples were freeze-dried, ground into powder (30 Hz, 1.5 min), and stored at −80 °C until needed. An amount of 50 mg of the powder was weighted and extracted with 0.5 mL methanol/water/hydrochloric acid (500:500:1, *v*/*v*/*v*). Then, the extract was vortexed for 5 min and treated by ultrasound for 5 min, and then centrifuged at 12,000× *g* under 4 °C for 3 min. The residue was re-extracted one more time by repeating the above steps under the same conditions. The supernatants were collected, and filtrated through a membrane filter (0.22 μm, Anpel) before LC-MS/MS analysis.

The sample extracts were analyzed using an UPLC-ESI-MS/MS system (UPLC, ExionLC™ AD, https://sciex.com.cn/ (accessed on 12 November 2021).; MS, Applied Biosystems 6500 Triple Quadrupole, https://sciex.com.cn/ (accessed on 12 November 2021)). The analytical conditions were as follows, UPLC: column, WatersACQUITY BEH C18 (1.7 µm, 2.1 mm*100 mm); solvent system, water (0.1% formic acid): methanol (0.1% formic acid); gradient program, 95:5 *v*/*v* at 0min, 50:50 *v*/*v* at 6 min, 5:95 *v*/*v* at 12 min, hold for 2 min, 95:5 *v*/*v* at 14 min; hold for 2min; flow rate, 0.35 mL/min; temperature, 40 °C; injection volume, 2 μL.

Linear ion trap (LIT) and triple quadrupole (QQQ) scans were acquired on a triple quadrupole-linear ion trap mass spectrometer (QTRAP), QTRAP^®^ 6500+ LC-MS/MS System, equipped with an ESI Turbo Ion-Spray interface, operating in positive ion mode, and controlled by Analyst 1.6.3 software (Sciex). The ESI source operation parameters were as follows: ion source, ESI+; source temperature, 550 °C; ion spray voltage (IS), 5500 V; curtain gas (CUR) pressure, 35 psi. Anthocyanins were analyzed using scheduled multiple reaction monitoring (MRM). Data acquisitions were performed using Analyst 1.6.3 software (Sciex). Multiquant 3.0.3 software (Sciex) was used to quantify all metabolites. Mass spectrometer parameters including the declustering potentials (DP) and collision energies (CE) for individual MRM transitions were established with further DP and CE optimization. A specific set of MRM transitions were monitored for each period according to the metabolites eluted within this period.

### 4.4. RNA Extraction and Sequencing

The petal samples with three biological replicates collected from X74 and *PF* were sent to Beijing Novogene bioinformatics technology Co., Ltd. for RNA-Seq. A total amount of 1 µg RNA per sample was used as input material for the RNA-seq library preparations. Sequencing libraries were generated using NEBNext^®^ UltraTM RNA Library Prep Kit for Illumina^®^ (NEB, San Diego, CA, USA) by following the manufacturer’s recommendations, and index codes were added to attribute sequences to each sample. The integrity and purity of the RNA samples were determined by 1% agarose gel electrophoresis, NanoPhotometer Spectrophotometer, and Agilent 2100 Bioanalyzer. The clustering of the index-coded samples was performed on a cBot Cluster Generation System using TruSeq PE Cluster Kit v3-cBot-HS (Illumia, San Diego, CA, USA) according to the manufacturer’s instructions. After cluster generation, the library preparations were sequenced on an Illumina Hiseq platform and 125–150 bp paired-end reads were generated.

### 4.5. Transcriptome Data Processing

Fastp v0.19.3 was used to filter the raw data, mainly to remove reads with adapters. Then, clean reads were obtained by removing reads containing adapter and ploy-N and low-quality reads. At the same time, the Q20, Q30, and GC contents of the clean data were calculated. All subsequent analyses were based on clean reads. The reference genome, *Gossypium hirsutum* (AD1) ‘TM-1’ genome ZJU-improved_v2.1_a1 [47], and its annotation files were downloaded from CottonGen (http://www.cottongen.org (accessed on 18 November 2021)). After indexing by using HISAT v2.1.0, the reference genome was used in alignment of the clean reads. FeatureCounts v1.6.2 was used to count the number of reads mapped to each gene, and to calculate the FPKM (the number of fragments per kilobase of transcript sequence per millions base pairs sequenced), which was used to characterize the abundance of gene transcripts [68].

### 4.6. Differential Gene Expression and Pathway Analysis

DESeq2 v1.22.1 was used to analyze the differential expression of genes between two groups, and the *p* value was corrected using the Benjamini & Hochberg method. The corrected *p* ≤ 0.05 and |log_2_(foldchange)| ≥ 1 were used as the thresholds for significantly differential expression. GO and KEGG enrichment analyses and visualization of DEGs were further implemented by employing the clusterProfiler R package and Sangerbox (http://vip.sangerbox.com/ (accessed on 22 November 2021)), respectively. The online iTAK was used to predict transcription factors (TFs) from all DEGs [69].

### 4.7. Gene Set Enrichment Analysis (GSEA)

All the expressed genes, regardless of whether or not they were differentially expressed from different comparison groups, were used for GSEA analysis, which was performed using the GSEA version 4.1.0 software with default parameters. Gene set enrichment analysis (GSEA) sorts all genes in the comparison group according to the multiple of difference between groups, and then analyzes the up and down regulation of the whole set according to the sorted results. The enrichment score for each gene set is then calculated using the entire ranked list, which reflects how the genes for each set are distributed in the ranked list. Normalized enriched score (NES) was determined for each gene set, which defines the degree of enrichment. The significantly enriched gene set was selected based on normalized enrichment score (NES) > 1 and false discovery rate (FDR) q-value ≤ 0.25.

### 4.8. Gene Network Construction and Visualization

A total of 5940 non-redundant DEGs from the four comparisons, WA_vs._PA, WB_vs._PB, PB_vs._PA, and WB_vs._WA, were selected for co-expression network analysis via the weighted gene co-expression network analysis (WGCNA) tools from the ImageGP platform (http://www.ehbio.com/Cloud_Platform/front/#/ (accessed on 10 December 2021)). The contents of seven representative metabolites (Procyanidin B2, Procyanidin C1, Cyanidin-3-O-glucoside, Delphinidin-3-O-glucoside, Pelargonidin-3-O-glucoside, Peonidin-3-O-galactoside, and Quercetin-3-O-glucoside) were used as the traits for WGCNA, and modules were obtained through WGCNA analysis with the default settings. Co-expression networks were constructed using the WGCNA (v1.29) package in R (Langfelder and Horvath, 2008). The modules were obtained using the automatic network construction function blockwiseModules with the default settings, except that the power was set to 18, TOM Type was signed, minModuleSize was set to 25, Deep split was set to 2, and mergeCutHeight was set to 0.25. The eigengene value was calculated for each module and used to test the association with each trait (metabolites). The networks were visualized using Cytoscape (v.3.0.0).

### 4.9. Quantitative Real-Time PCR Analysis

Each sample was ground to powder in liquid nitrogen and 0.1 g power was used to extract total RNA using an RNA extraction kit (Huayueyang Biotechnology Inc., Beijing, China) by following the specifications of the kit. The cDNA was synthesized by using an EASYspin Plus Plant RNA Kit (Aidlab Biotechnologies Co., Ltd.; Beijing, China) from 1 μg of total RNA, according to the manufacturer’s instructions. Three biological and three technical replicates for each reaction were analyzed on a LightCycler^®^ 480 instrument (Roche, Switzerland) with a first step of 95 °C for 5 min, followed by 45 cycles of 95 °C for 10 s, 60 °C for 15 s, and 72 °C for 20 s. The gene expression levels were calculated with the 2^−ΔΔcT^ method. The cotton GhUBQ7 gene (DQ116441) was amplified as an internal control gene. All primers are listed in Appendix A.

## 5. Conclusions

In this study, a total of 38 anthocyanin metabolites were identified in the white and pink petals. Among them, Quercetin-3-O-glucoside and Delphinidin-3-O-glucoside are abundant in both white and pink petals. Cyanidin-3-O-glucoside, with a markedly higher abundance in pink petals, may be the major contributor to the formation of pink petals. Other glycosidic derivatives of cyanidins, pelargonidins, and peonidins that were not detected in white petals may also contribute to the intensity of pink color. The combined analysis of RNA-seq and metabolome revealed that the expression of *MYB21*, *UGT88F3*, *GSTF12*, and *VPS32.3* was associated with the accumulation of Cyanidin-3-O-glucoside in petals. In addition, many of the 78 genes identified through WGCNA were correlated with Cyanidin-3-O-glucoside and Delphinidin-3-O-glucoside. While the relationship between these genes and anthocyanin biosynthesis needs further investigation, we hope that our work has contributed to the understanding of anthocyanin biosynthesis in upland cotton.

## Figures and Tables

**Figure 1 ijms-23-10137-f001:**
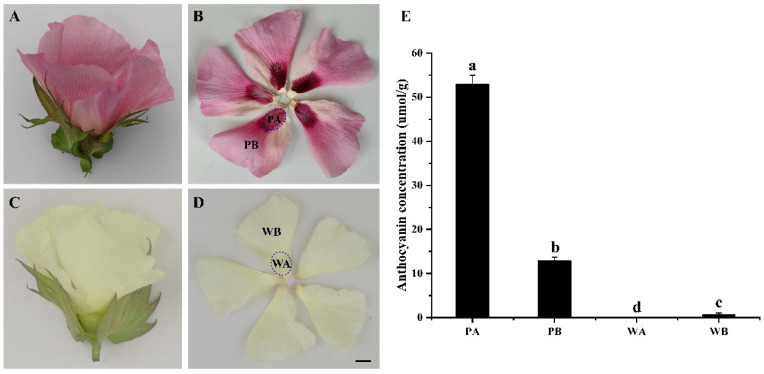
The color phenotypes characterization and total anthocyanin content of different parts of petals of varieties XinLuZao 74 (X74) and *Pink Flower* (*PF*). (**A**,**B**) Phenotypes of *PF* mutant petals. (**C**,**D**) Phenotypes of X74 petals. (**E**) Total anthocyanin concentration in different regions of X74 and *PF* petals. All data are shown as mean ± SE (*n* = 3). Different letters represented statistically significant differences (one-way ANOVA, *p* < 0.05). Scale bar: 1 cm. PA, PB, WA, and WB represent the spot region and non-spot region of *PF* and X74 petals, respectively.

**Figure 2 ijms-23-10137-f002:**
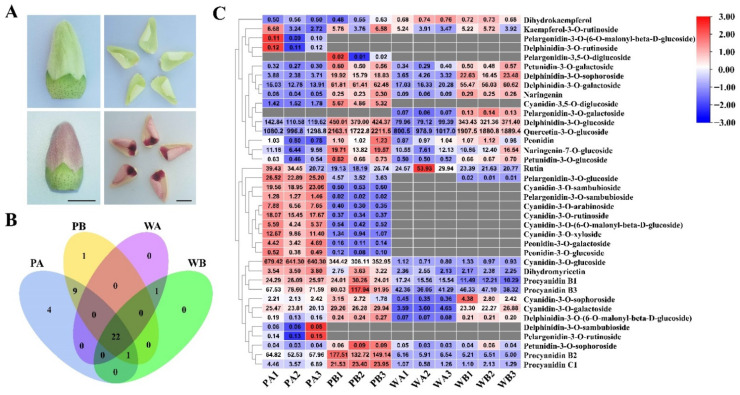
Phenotypes of XinLuZao 74 (X74) and *Pink Flower* (*PF*), and preliminary analysis of metabolomics data. (**A**) Sampling period for metabolite analysis and RNA-sequencing (RNA-Seq) analysis. Scale bar: 1 cm. (**B**) Venn diagram of metabolites accumulated in different regions of petals of X74 and *PF*. (**C**) Cluster heat map of the 38 anthocyanin metabolites in the X74 and *PF* cultivars. Intensity values were adjusted by log transformation and then normalized. PA, PB, WA, and WB represent the spot region and non-spot region of *PF* and X74 petals, respectively. PA1/PA2/PA3, PB1/PB2/PB3, WA1/WA2/WA3, and WB1/WB2/WB3 represents three biological replicates for each sample, respectively.

**Figure 3 ijms-23-10137-f003:**
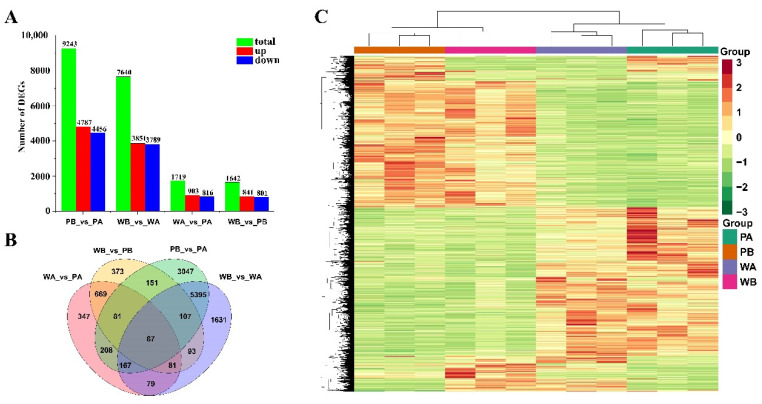
Differential genes (DEGs) analysis of petals of different colors and regions. (**A**) Numbers of DEGs in pairwise comparisons of the four libraries. (**B**) Venn diagram showing DEG distributions. (**C**) Expression profile clustering of different samples. The color scale on the right represents re-processed log10 (FPKM+1) using heatmap, representing the relative expression level. The expression variance for each gene is indicated by different colors ranging from low (green) to high (red). PA, PB, WA, and WB represent the spot region and non-spot region of *PF* and X74 petals, respectively.

**Figure 4 ijms-23-10137-f004:**
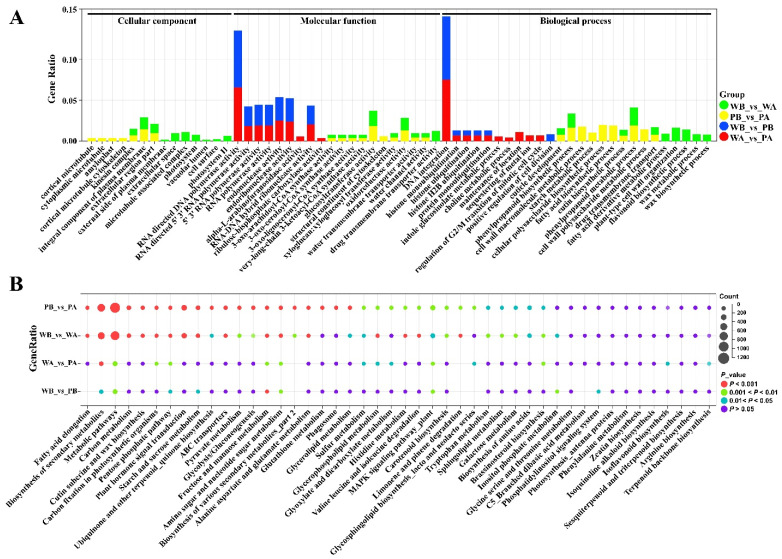
GO and KEGG enrichment analysis of all DEGs. (**A**) Enrichment of the top 10 GO pathways of all DEGs according to the *p*-value < 0.05. (**B**) Enrichment of the top 20 KEGG pathways of all DEGs. The red represents significant enrichment.

**Figure 5 ijms-23-10137-f005:**
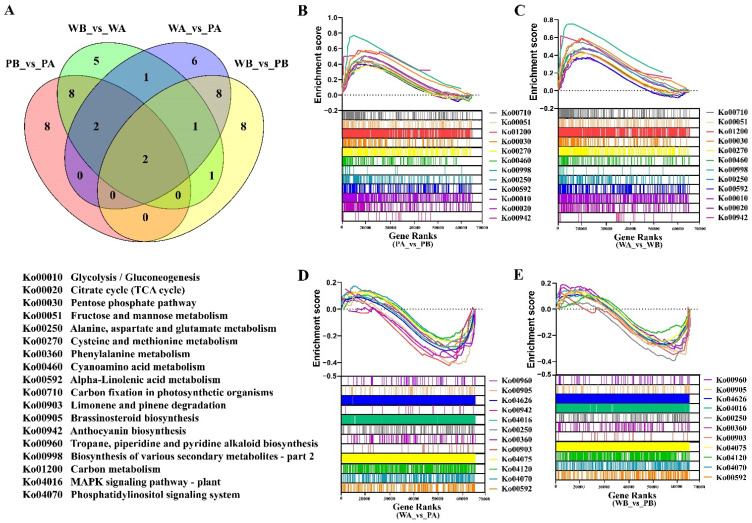
Gene set enrichment analysis (GSEA) was performed in four different comparison groups. (**A**) Venn diagram shows the distribution of the top20 enriched gene sets from four comparison groups (PA_vs._PB, WA_vs._WB, WA_vs._PA, and WB_vs._PB). (**B**,**C**) Represents the overlapping enriched gene sets from GSEA, comparing spot with non-spot of petals of *PF* or X74 (PA_vs._PB and WA_vs._WB). (**D**,**E**) Represents the overlapping enriched gene sets from GSEA comparing *PF* petals colored with X74 petals colorless regions of spot or non-spot. (WA_vs._PA and WB_vs._PB). PA, PB, WA, and WB represent the spot region and non-spot region of *PF* and X74 petals, respectively.

**Figure 6 ijms-23-10137-f006:**
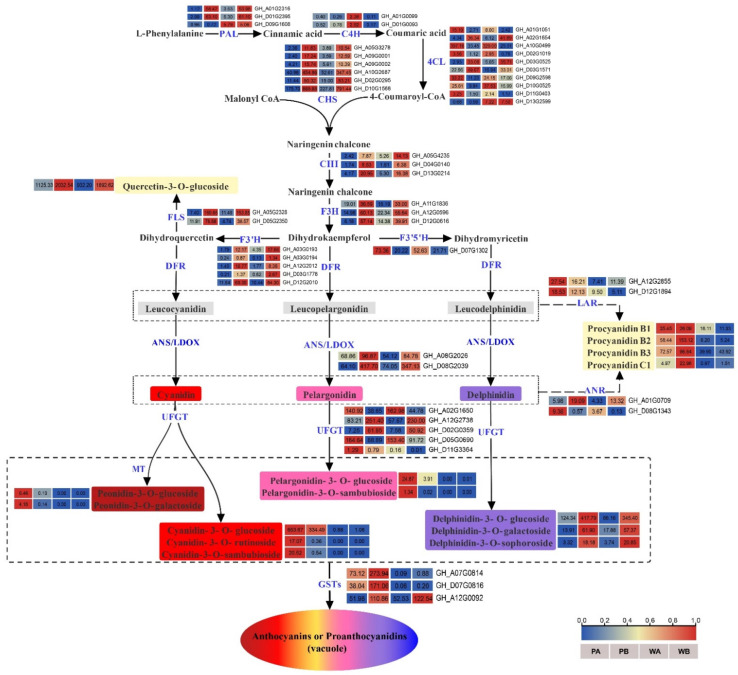
The pathway related to anthocyanin biosynthesis and the heat map of changes in metabolites and transcripts in anthocyanin metabolisms. The expression levels of Phenylalanine ammonia lyase (PAL), Cinnamate 4-hydroxylase (C4H), and 4-coumarin: Co-A ligase (4CL), Chalcone synthase (CHS), Chalcone isomerase (CHI), Flavanone 3-hydroxylase (F3H), Flavonoid 3′-hydroxylase (F3′H), Flavonoid 3′,5′-hydroxylase (F3′5′H), Dihydroflavonol 4-reductase (DFR), Anthocyanidin synthase (ANS), Leucoanthocyanidin reductase (LAR), Anthocyanidin reductase (ANR), Flavonoid 3-O-glucosyltransferase (UFGT), Glutathione S-transferase (GST), and anthocyanin metabolites are shown.

**Figure 7 ijms-23-10137-f007:**
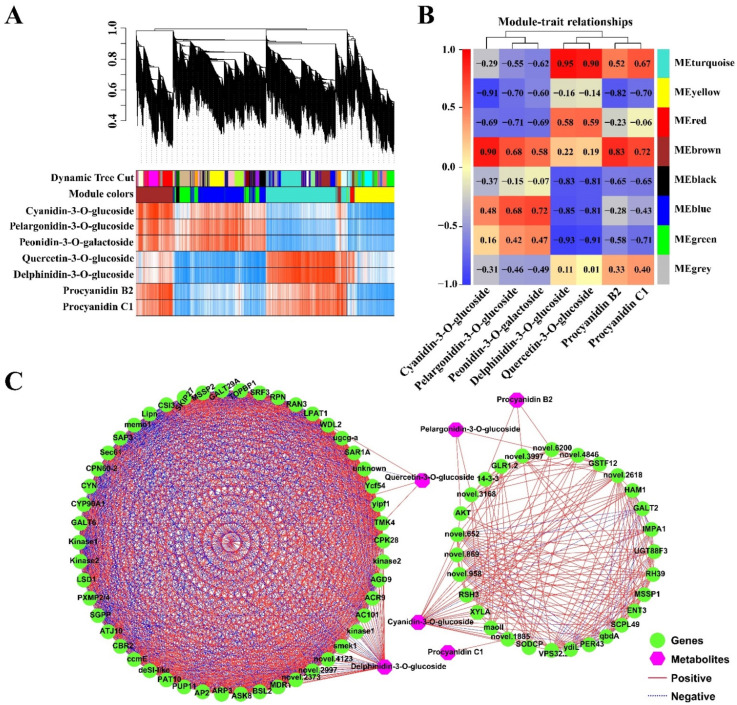
Weighted gene co-expression network analysis (WGCNA) of DEGs (with 7 metabolites, |PCC| ≥ 0.95) identified in pink and white petals. (**A**) Hierarchical clustering tree (cluster dendrogram) results showed eight expression modules, labeled with different colors. (**B**) Module–anthocyanin relationship analysis. The value inside each box represents Pearson’s correlation coefficient between the module with anthocyanin. The color scale on the left represents the degree of correlation between modules and anthocyanins and the red represent high correlation. (**C**) Connection network between DEGs and anthocyanin metabolites; the green ellipsoid represents DEGs; the pink diamond represents the metabolites. The red line indicates a positive correlation, and the blue dotted line indicates a negative correlation.

## Data Availability

The original contributions presented in the study are publicly available. This data can be found here: National Center for Biotechnology Information (NCBI) BioProject database under accession number PRJNA825010 (Release date: 1 October 2022).

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
