# Peer review of "Comparative Metabolome and Transcriptome Analysis of Anthocyanin Biosynthesis in White and Pink Petals of Cotton (*Gossypium hirsutum* L.)"

_ijms, 2022, doi:10.3390/ijms231710137_

Round 1
Reviewer 1 Report
The manuscript by Shao et al. describes the study of revealing the key genes regulating the bio synthesis and accumulation of anthocyanins by comparison of gene expression of the white and pink petals in Upland cotton using RNA-seq analysis. It is interesting to identify the major genes by looking at the regulatory genes and metabolites associated with anthocyanin biosynthesis pathway.
However, this research only confirmed the difference in gene expression through RNA-seq data analysis, and there wasn’t biological validation for the identified genes, so I could only know the candidate genes and I don’t have confidence whether the genes directly affect the anthocyanin synthesis.
Overall, the size of the figures was too small compared to the contents and the resolution was low, so it was difficult to read the gene names or numbers on the figures even if the PDF file was enlarged. It should be changed high resolution figures and increase font size to make the contents clear.
The introduction was very well explained and easy to understand. Except absence of biological validation data, the composition of the experiments, the research methods, and the presentation of the results look well done. In particular, it is very interesting to find candidate key genes related to biosynthesis and transport in anthocyanidins in cotton petal, but it is disappointing that these genes haven’t confirmed whether they function actually or not.
The following comments could help to further improve the quality of the paper.
1) Figure 1A~1D : In the pink petal, PA and PB are clearly different. So, it seems appropriate to distinguish them. But in the white petal, there is no difference at all. Is it meaningful to only positionally distinguish WA and WB like the pink petal?
2) Figure 1E : What is the meaning of a~d on the bars?
3) In Figure 4A and 4B, you used different fonts for them. Also, I can’t read almost contents because low resolution of figures.
4) Line 193 : ‘Glutathione metabolism. -> ‘Glutathione metabolism’.
5) Table 6 : Did you not put the contexts for ‘GS DETAILS’ on table? I think you should fill them about their function.
6) Line 210~212 : You found some other genes when you used GSEA analysis compare to KEGG analysis. What are the reasons for these different results? It would be nice to explain it in discussion.
7) Line 538~546 : Unlike other method parts, only this part has a different font. Use same font.
Reviewer 2 Report
This manuscript is well written and the results clearly presented. Experimental design appears good and adequate explanation of methods used. References cited provide good information and are not excessive, well balanced. The conclusions are at the correct level with the data, and do not over-step their significance. Overall a good manuscript, I would recommend to accept this manuscript for publication.
Round 2
Reviewer 1 Report
I checked the responses to the comments. Most of them explained well enough to understand, but I still think that the part about biological verification data is lacking. But aside from that, it's an interesting study overall. I hope to proceed with the follow-up experiments.
The quality of the figures has been improved a lot, but there are still many parts they looked too small and blurry. It would be better to raise the higher resolution, and if possible, it would be better to enlarge the figure size to make it easier to see.
Author Response
Dear Reviewer #1,
We appreciate you for your precious time in reviewing our paper and providing valuable comments, which are valuable in improving the quality of our manuscript. Below is our point-by-point response to the referee’s comments.
Best regards
All authors
Revision - authors’ response:
Reviewer #1
I checked the responses to the comments. Most of them explained well enough to understand, but I still think that the part about biological verification data is lacking. But aside from that, it's an interesting study overall. I hope to proceed with the follow-up experiments.
The quality of the figures has been improved a lot, but there are still many parts they looked too small and blurry. It would be better to raise the higher resolution, and if possible, it would be better to enlarge the figure size to make it easier to see.
Response to Reviewer #1
Minor comments:
Comment 1. I checked the responses to the comments. Most of them explained well enough to understand, but I still think that the part about biological verification data is lacking. But aside from that, it's an interesting study overall. I hope to proceed with the follow-up experiments.
Response: Thanks for your comment. First of all, the main focus of this study is to reveal the regulation mechanism of anthocyanin synthesis in white and pink petals of Upland cotton through transcriptome and metabolome. The current research content has basically reached our expected results. More importantly, due to the impact of COVID-19 pandemic, the laboratory is not open and experimental work is not allowed. we hope the reviewer #1 understand. Finally, we are very grateful for the comments of the reviewers. We will follow your guidance and complete the biological verification in the follow-up study.
Comment 2. The quality of the figures has been improved a lot, but there are still many parts they looked too small and blurry. It would be better to raise the higher resolution, and if possible, it would be better to enlarge the figure size to make it easier to see.
Response: Thanks for your kind reminders. As suggested by the reviewer, we rechecked the quality of all the pictures, and then we improved the resolution and size of Figures 4, 6 and 7.